# Sensory Integration and Density Estimation

**Joseph G. Makin and Philip N. Sabes**
Center for Integrative Neuroscience/Department of Physiology
University of California, San Francisco

San Francisco, CA 94143-0444 USA
makin, sabes @phy.ucsf.edu

## Abstract

The integration of partially redundant information from multiple sensors is a standard computational problem for agents interacting with the world. In man and other primates, integration has been shown psychophysically to be nearly optimal in the sense of error minimization. An influential generalization of this notion of optimality is that populations of multisensory neurons should retain all the information from their unisensory afferents about the underlying, common stimulus [1]. More recently, it was shown empirically that a neural network trained to perform latent-variable density estimation, with the activities of the unisensory neurons as observed data, satisfies the information-preservation criterion, even though the model architecture was not designed to match the true generative process for the data [2]. We prove here an analytical connection between these seemingly different tasks, density estimation and sensory integration; that the former implies the latter for the model used in [2]; but that this does not appear to be true for all models.

## 1 Introduction

A sensible criterion for integration of partially redundant information from multiple senses is that no information about the underlying cause be lost. That is, the multisensory representation should contain all of the information about the stimulus as the unisensory representations together did. A variant on this criterion was first proposed in [1]. When satisfied, and given sensory cues that have been corrupted with Gaussian noise, the most probable multisensory estimate of the underlying stimulus property (height, location, etc.) will be a convex combination of the estimates derived independently from the unisensory cues, with the weights determined by the variances of the corrupting noise—as observed psychophysically in monkey and man, e.g., [3, 4].

The task of plastic organisms placed in novel environments is to learn, from scratch, how to perform this task. One recent proposal [2] is that primates treat the activities of the unisensory populations of neurons as observed data for a latent-variable density-estimation problem. Thus the activities of a population of multisensory neurons play the role of latent variables, and the model is trained to generate the same distribution of unisensory activities when they are driven by the multisensory neurons as when they are driven by their true causes in the world. The idea is that the latent variables in the model will therefore come to correspond (in some way) to the latent variables that truly underlie the observed distribution of unisensory activities, including the structure of correlations across populations. Then it is plausible to suppose that, for any particular value of the stimulus, inference to the latent variables of the model is "as good as" inference to the true latent cause, and that therefore the information criterion will be satisfied. Makin et alia showed precisely this, empirically, using an exponential-family harmonium (a generalization of the restricted Boltzmann machine [5]) as the density estimator [2].

Here we prove analytically that successful density estimation in certain models, including that of [2], will necessarily satisfy the information-retention criterion. In variant architectures, the guarantee does not hold.

## 2 Theoretical background

### 2.1 Multisensory integration and information retention

Psychophysical studies have shown that, when presented with cues of varying reliability in two different sense modalities but about a common stimulus property (e.g., location or height), primates (including humans) estimate the property as a convex combination of the estimates derived independently from the unisensory cues, where the weight on each estimate is proportional to its reliability [3, 4]. Cue reliability is measured as the inverse variance in performance across repeated instances of the unisensory cue, and will itself vary with the amount of corrupting noise (e.g., visually blur) added to the cue. This integration rule is optimal in that it minimizes error variance across trials, at least for Gaussian corrupting noise.

Alternatively, it can be seen as a special case of a more general scheme [6]. Assuming a uniform prior distribution of stimuli, the optimal combination just described is equal to the peak of the posterior distribution over the stimulus, conditioned on the noisy cues $(\mathbf{y}_1, \mathbf{y}_2)$:

$$\mathbf{x}^* = \underset{\mathbf{x}}{\operatorname{argmax}} \Pr[\mathbf{X} = \mathbf{x} | \mathbf{y}_1, \mathbf{y}_2].$$

For Gaussian corrupting noise, this posterior distribution will itself be Gaussian; but even for integration problems that yield non-Gaussian posteriors, humans have been shown to estimate the stimulus with the peak of that posterior [7].

This can be seen as a consequence of a scheme more general still, namely, encoding not merely the peak of the posterior, but the entire distribution [1, 8]. Suppose again, for simplicity, that $\Pr[\mathbf{X} | \mathbf{Y}_1, \mathbf{Y}_2]$ is Gaussian. Then if $\mathbf{x}^*$ is itself to be combined with some third cue $(\mathbf{y}_3)$, optimality requires keeping the variance of this posterior as well, since it (along with the reliability of $\mathbf{y}_3$) determines the weight given to $\mathbf{x}^*$ in this new combination. This scheme is especially relevant when $\mathbf{y}_1$ and $\mathbf{y}_2$ are not "cues" but the activities of populations of neurons, e.g. visual and auditory, respectively. Since sensory information is more likely to be integrated in the brain in a staged, hierarchical fashion than in a single common pool [9], optimality requires encoding at least the first two cumulants of the posterior distribution. For more general, non-Gaussian posteriors, the entire posterior should be encoded [1, 6]. This amounts [1] to requiring, for downstream, "multisensory" neurons with activities $\mathbf{Z}$, that:

$$\Pr[\mathbf{X} | \mathbf{Z}] = \Pr[\mathbf{X} | \mathbf{Y}_1, \mathbf{Y}_2].$$

When information about $\mathbf{X}$ reaches $\mathbf{Z}$ *only* via $\mathbf{Y} = [\mathbf{Y}_1, \mathbf{Y}_2]$ (i.e., $\mathbf{X} \to \mathbf{Y} \to \mathbf{Z}$ forms a Markov chain), this is equivalent (see Appendix) to requiring that no information about the stimulus be lost in transforming the unisensory representations into a multisensory representation; that is,

$$\mathcal{I}(\mathbf{X}; \mathbf{Z}) = \mathcal{I}(\mathbf{X}; \mathbf{Y}),$$

where $\mathcal{I}(A; B)$ is the mutual information between $A$ and $B$.

Of course, if there is any noise in the transition from unisensory to multisensory neurons, this equation cannot be satisfied exactly. A sensible modification is to require that this noise be the *only* source of information loss. This amounts to requiring that the information equality hold, not for $\mathbf{Z}$, but for any set of sufficient statistics for $\mathbf{Z}$ as a function of $\mathbf{Y}$, $\mathbf{T}_\mathbf{z}(\mathbf{Y})$; that is,

$$\mathcal{I}(\mathbf{X}; \mathbf{T}_\mathbf{z}(\mathbf{Y})) = \mathcal{I}(\mathbf{X}; \mathbf{Y}). \tag{1}$$

### 2.2 Information retention and density estimation

A rather general statement of the role of neural sensory processing, sometimes credited to Helmholtz, is to make inferences about states of affairs in the world, given only the data supplied by the sense organs. Inference is hard because the mapping from the world's states to sense data is

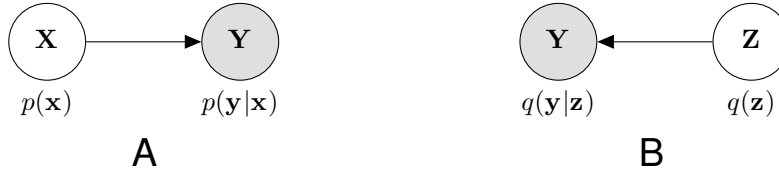

Figure 1: Probabilistic graphical models. (A) The world's generative process. (B) The model's generative process. Observed nodes are shaded. After training the model ($q$), the marginals match: $p(\mathbf{y}) = q(\mathbf{y})$.

not invertible, due both to noise and to the non-injectivity of physical processes (as in occlusion). A powerful approach to this problem used in machine learning, and arguably by the brain [10, 11], is to build a generative model for the data ($\mathbf{Y}$), including the influence of unobserved (latent) variables ($\mathbf{Z}$). The latent variables at the top of a hierarchy of such models would presumably be proxies for the *true* causes, states of affairs in the world ($\mathbf{X}$).

In density estimation, however, the objective function for learning the parameters of the model is that:

$$\int_{\mathbf{x}} p(\mathbf{y}|\mathbf{x})p(\mathbf{x})d\mathbf{x} = \int_{\mathbf{z}} q(\mathbf{y}|\mathbf{z})q(\mathbf{z})d\mathbf{z} \qquad (2)$$

(Fig. 1), i.e., that the "data distribution" of $\mathbf{Y}$ match the "model distribution" of $\mathbf{Y}$; and this is consistent with models that throw away information about the world in the transformation from observed to latent variables, or even to their sufficient statistics. For example, suppose that the world's generative process looked like this:

**Example 2.1.** The prior $p(x)$ is the flip of an unbiased coin; and the emission $p(y|x)$ draws from a standard normal distribution, takes the absolute value of the result, and then multiplies by $-1$ for tails and $+1$ for heads. Information about the state of $X$ is therefore perfectly represented in $Y$. But a trained density-estimation model with, say, a Gaussian emission model, $q(y|\mathbf{z})$, would not bother to encode any information in $\mathbf{Z}$, since the emission model alone can represent all the data (which just look like samples from a standard normal distribution). Thus $Y$ and $\mathbf{Z}$ would be independent, and Eq. 1 would not be satisfied, even though Eq. 2 would.

This case is arguably pathological, but similar considerations apply for more subtle variants. In addition to Eq. 2, then, we shall assume something more: namely, that the "noise models" for the world and model match; i.e., that $q(\mathbf{y}|\mathbf{z})$ has the same functional form as $p(\mathbf{y}|\mathbf{x})$. More precisely, we assume:

$$\exists \text{ functions } f(\mathbf{y};\boldsymbol{\lambda}), \phi(\mathbf{x}), \psi(\mathbf{z}) \ni \qquad \begin{aligned} p(\mathbf{y}|\mathbf{x}) &= f\big(\mathbf{y};\phi(\mathbf{x})\big), \\ q(\mathbf{y}|\mathbf{z}) &= f\big(\mathbf{y};\psi(\mathbf{z})\big). \end{aligned} \qquad (3)$$

In [2], for example, $f(\mathbf{y};\boldsymbol{\lambda})$ was assumed to be a product of Poisson distributions, so the "proximate causes" $\boldsymbol{\Lambda}$ were a vector of means. Note that the functions $\phi(\mathbf{x})$ and $\psi(\mathbf{z})$ induce distributions over $\boldsymbol{\Lambda}$ which we shall call $p(\boldsymbol{\lambda})$ and $q(\boldsymbol{\lambda})$, respectively; and that:

$$\mathbb{E}_{p(\boldsymbol{\lambda})}[f(\mathbf{y};\boldsymbol{\lambda})] = \mathbb{E}_{p(\mathbf{x})}[f(\mathbf{y};\phi(\mathbf{x})] = \mathbb{E}_{q(\mathbf{z})}[f(\mathbf{y};\psi(\mathbf{z})] = \mathbb{E}_{q(\boldsymbol{\lambda})}[f(\mathbf{y};\boldsymbol{\lambda})], \qquad (4)$$

where the first and last equalities follows from the "law of the unconscious statistician," and the second follows from Eqs. 2 and 3.

## 3 Latent-variable density estimation for multisensory integration

In its most general form, the aim is to show that Eq. 4 implies, perhaps with some other constraints, Eq. 1. More concretely, suppose the random variables $\mathbf{Y}_1, \mathbf{Y}_2$, provided by sense modalities 1 and 2, correspond to noisy observations of an underlying stimulus. These could be noisy cues, but they could also be the activities of populations of neurons (visual and proprioceptive, say, for concreteness). Then suppose a latent-variable density estimator is trained on these data, until it assigns the same probability, $q(\mathbf{y}_1, \mathbf{y}_2)$, to realizations of the observations, $[\mathbf{y}_1, \mathbf{y}_2]$, as that with which they appear, $p(\mathbf{y}_1, \mathbf{y}_2)$. Then we should like to know that inference to the latent variables in the model,

i.e., computation of the sufficient statistics $\mathbf{T_z}(\mathbf{Y}_1, \mathbf{Y}_2)$, throws away no information about the stimulus. In [2], where this was shown empirically, the density estimator was a neural network, and its latent variables were interpreted as the activities of downstream, multisensory neurons. Thus the transformation from unisensory to multisensory representation was shown, after training, to obey this information-retention criterion.

It might seem that we have already assembled sufficient conditions. In particular, knowing that the "noise models match," Eq. 3, might seem to guarantee that the data distribution and model distribution have the same sufficient statistics, since sufficient statistics depend only on the form of the conditional distribution. Then $\mathbf{T_z}(\mathbf{Y})$ would be sufficient for $\mathbf{X}$ as well as for $\mathbf{Z}$, and the proof complete. But this sense of "form of the conditional distribution" is stronger than Eq. 4. If, for example, the image of $\mathbf{z}$ under $\psi(\cdot)$ is lower-dimensional than the image of $\mathbf{x}$ under $\phi(\cdot)$, then the conditionals in Eq. 3 will have different forms as far as their sufficient statistics go. An example will clarify the point.

**Example 3.1.** Let $p(y)$ be a two-component mixture of a (univariate) Bernoulli distribution. In particular, let $\phi(x)$ and $\psi(z)$ be the identity maps, $\Lambda$ provide the mean of the Bernoulli, and $p(X = 0.4) = 1/2$, $p(X = 0.6) = 1/2$. The mixture marginal is therefore another Bernoulli random variable, with equal probability of being 1 or 0. Now consider the "mixture" model $q$ that has the same noise model, i.e., a univariate Bernoulli distribution, but a prior with all its mass at a single mixing weight. If $q(Z = 0.5) = 1$, this model will satisfy Eq. 4. But a (minimal) sufficient statistic for the latent variables under $p$ is simply the single sample, $y$, whereas the minimal sufficient statistic for the latent variable under $q$ is the nullset: the observation tells us nothing about $Z$ because it is always the same value.

To rule out such cases, we propose (below) further constraints.

## 3.1 Proof strategy

We start by noting that any sufficient statistics $\mathbf{T_z}(\mathbf{Y})$ for $\mathbf{Z}$ are also sufficient statistics for any function of $\mathbf{Z}$, since all the information about the output of that function must pass through $\mathbf{Z}$ first (Fig. 2A). In particular, then, $\mathbf{T_z}(\mathbf{Y})$ are sufficient statistics for the proximate causes, $\boldsymbol{\Lambda} = \psi(\mathbf{Z})$. That is, for any $\boldsymbol{\lambda}$ generated by the model, Fig. 1B, $\mathbf{t_z}(\mathbf{y})$ for the corresponding $\mathbf{y}$ drawn from $f(\mathbf{y}; \boldsymbol{\lambda})$ are sufficient statistics. What about the $\boldsymbol{\lambda}$ generated by the world, Fig. 1A? We should like to show that $\mathbf{t_z}(\mathbf{y})$ are sufficient for them as well. This will be the case if, for every $\boldsymbol{\lambda}$ produced by the world, there exists a vector $\mathbf{z}$ such that $\psi(\mathbf{z}) = \boldsymbol{\lambda}$.

This minimal condition is hard to prove. Instead we might show a slightly stronger condition, that $(q(\boldsymbol{\lambda}) = 0) \implies (p(\boldsymbol{\lambda}) = 0)$, i.e., to any $\boldsymbol{\lambda}$ that can be generated by the world, the model assigns nonzero probability. This is sufficient (although unnecessary) for the existence of a vector $\mathbf{z}$ for every $\boldsymbol{\lambda}$ produced by the world. Or we might pursue a stronger condition still, that to any $\boldsymbol{\lambda}$ that can be generated by the world, the model and data assign *the same* probability: $q(\boldsymbol{\lambda}) = p(\boldsymbol{\lambda})$. If one considers the marginals $p(\mathbf{y}) = q(\mathbf{y})$ to be mixture models, then this last condition is called the "identifiability" of the mixture [12], and for many conditional distributions $f(\mathbf{y}; \boldsymbol{\lambda})$, identifiability conditions have been proven. Note that mixture identifiability is taken to be a property of the conditional distribution, $f(\mathbf{y}; \boldsymbol{\lambda})$, not the marginal, $p(\mathbf{y})$; so, e.g., without further restriction, a mixture model is not identifiable even if there exist just two prior distributions, $p_1(\boldsymbol{\lambda})$, $p_2(\boldsymbol{\lambda})$, that produce identical marginal distributions.

To see that identifiability, although sufficient (see below) is unnecessary, consider again the two-component mixture of a (univariate) Bernoulli distribution:

**Example 3.2.** Let $p(X = 0.4) = 1/2$, $p(X = 0.6) = 1/2$. If the model, $q(y|z)q(z)$, has the same form, but mixing weights $q(Z = 0.3) = 1/2, q(Z = 0.7) = 1/2$, its mixture marginal will match the data distribution; yet $p(\lambda) \neq q(\lambda)$, so the model is clearly unidentifiable. Nevertheless, the sample itself, $y$, is a (minimal) sufficient statistic for both the model and the data distribution, so the information-retention criterion will be satisfied.

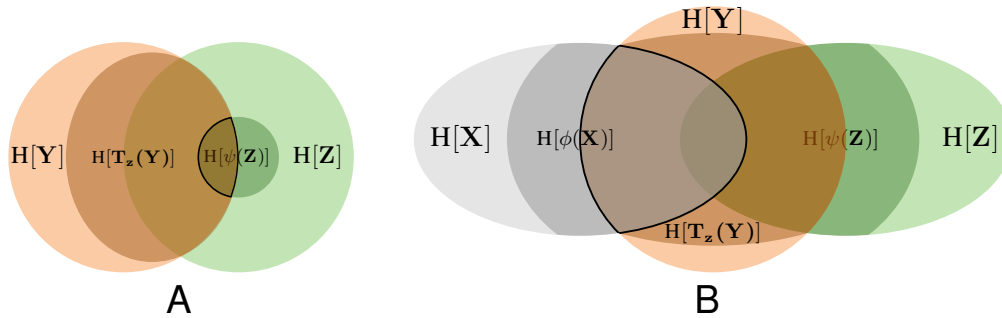

Figure 2: Venn diagrams for information. (A) $\psi(\mathbf{Z})$ being a deterministic function of $\mathbf{Z}$, its entropy (dark green) is a subset of the latter's (green). The same is true for the entropies of $\mathbf{T_z}(\mathbf{Y})$ (dark orange) and $\mathbf{Y}$ (orange), but additionally their intersections with $H[\mathbf{Z}]$ are identical because $\mathbf{T_z}$ is a sufficient statistic for $\mathbf{Z}$. The mutual information values $\mathcal{I}(\psi(\mathbf{Z}); \mathbf{Y})$ and $\mathcal{I}(\psi(\mathbf{Z}); \mathbf{T_z}(\mathbf{Y}))$ (i.e., the intersections of the entropies) are clearly identical (outlined patch). This corresponds to the derivation of Eq. 6. (B) When $\psi(\mathbf{Z})$ is a sufficient statistic for $\mathbf{Y}$, as guaranteed by Eq. 3, the intersection of its entropy with $H[\mathbf{Y}]$ is the same as the intersection of $H[\mathbf{Z}]$ with $H[\mathbf{Y}]$; likewise for $H[\phi(\mathbf{X})]$ and $H[\mathbf{X}]$ with $H[\mathbf{Y}]$. Since all information about $\mathbf{X}$ reaches $\mathbf{Z}$ via $\mathbf{Y}$, the entropies of $\mathbf{X}$ and $\mathbf{Z}$ overlap only on $H[\mathbf{Y}]$. Finally, if $p(\phi(\mathbf{x})) = q(\psi(\mathbf{z}))$, and $\Pr[\mathbf{Y}|\phi(\mathbf{X})] = \Pr[\mathbf{Y}|\psi(\mathbf{Z})]$ (Eq. 3), then the entropies of $\phi(\mathbf{X})$ and $\psi(\mathbf{Z})$ have the same-sized overlaps (but not the same overlaps) with $H[\mathbf{Y}]$ and $H[\mathbf{T_z}(\mathbf{Y})]$. This guarantees that $\mathcal{I}(\mathbf{X}; \mathbf{T_z}(\mathbf{Y})) = \mathcal{I}(\mathbf{X}; \mathbf{Y})$ (see Eq. 7).

In what follows we shall assume that the mixtures are finite. This is the case when the model is an exponential-family harmonium (EFH)[1], as in [2]: there are at most $K := 2^{|\text{hiddens}|}$ mixture components. It is not true for real-valued stimuli $\mathbf{X}$. For simplicity, we shall nevertheless assume that there are at most $2^{|\text{hiddens}|}$ configurations of $\mathbf{X}$ since: (1) the stimulus must be discretized immediately upon transduction by the nervous system, the brain (presumably) having only finite representational capacity; and (2) if there were an infinite number of configurations, Eq. 2 could not be satisfied exactly anyway. Eq. 4 can therefore be expressed as:

$$\sum_i^I f(\mathbf{y}; \boldsymbol{\lambda}) p(\boldsymbol{\lambda}) = \sum_j^J f(\mathbf{y}; \boldsymbol{\lambda}) q(\boldsymbol{\lambda}), \tag{5}$$

where $I \le K, J \le K$.

### 3.2 Formal description of the model, assumptions, and result

- **The general probabilistic model.** This is given by the graphical models in Fig. 1. "The world" generates data according to Fig. 1A ("data distribution"), and "the brain" uses Fig. 1B. None of the distributions labeled in the diagram need be equal to each other, and in fact $\mathbf{X}$ and $\mathbf{Z}$ may have different support.

- **The assumptions.**
  1. The noise models "match": Eq. 3.
  2. The number of hidden-variable states is finite, but otherwise arbitrarily large.
  3. Density estimation has been successful; i.e., the data and model marginals over $\mathbf{Y}$ match: Eq. 2
  4. The noise model/conditional distribution $f(\mathbf{y}; \boldsymbol{\lambda})$ is identifiable: if $p(\mathbf{y}) = q(\mathbf{y})$, then $p(\boldsymbol{\lambda}) = q(\boldsymbol{\lambda})$. This condition holds for a very broad class of distributions.

- **The main result.** Information about the stimulus is retained in inferring the latent variables of the model, i.e. in the "feedforward" ($\mathbf{Y} \rightarrow \mathbf{Z}$) pass of the model. More precisely,

information loss is due only to noise in the hidden layer (which is unavoidable), not to a failure of the inference procedure: Eq. 1.

More succinctly: for identifiable mixture models, Eq. 5 and Eq. 3 together imply Eq. 1.

### 3.3 Proof

First, for any set of sufficient statistics $\mathbf{T_z}(\mathbf{Y})$ for $\mathbf{Z}$:

$$
\begin{aligned}
\mathcal{I}(\mathbf{Y};\psi(\mathbf{Z})|\mathbf{T_z}(\mathbf{Y})) &\leq \mathcal{I}(\mathbf{Y};\mathbf{Z}|\mathbf{T_z}(\mathbf{Y})) && \text{data-processing inequality [13]} \\
&= 0 && \mathbf{T_z}(\mathbf{Y}) \text{ are sufficient for } \mathbf{Z} \\
\implies 0 &= \mathcal{I}(\mathbf{Y};\psi(\mathbf{Z})|\mathbf{T_z}(\mathbf{Y})) && \text{Gibbs's inequality} \\
&= \mathrm{H}[\psi(\mathbf{Z})|\mathbf{T_z}(\mathbf{Y})] - \mathrm{H}[\psi(\mathbf{Z})|\mathbf{Y},\mathbf{T_z}(\mathbf{Y})] && \text{def'n cond. mutual info.} \\
&= \mathrm{H}[\psi(\mathbf{Z})|\mathbf{T_z}(\mathbf{Y})] - \mathrm{H}[\psi(\mathbf{Z})|\mathbf{Y}] && \mathbf{T_z}(\mathbf{Y}) \text{ deterministic} \\
&\quad - \mathrm{H}[\psi(\mathbf{Z})] + \mathrm{H}[\psi(\mathbf{Z})] && = 0 \\
\implies \mathcal{I}(\psi(\mathbf{Z});\mathbf{T_z}(\mathbf{Y})) &= \mathcal{I}(\psi(\mathbf{Z});\mathbf{Y}). && \text{def'n mutual info.}
\end{aligned}
\tag{6}
$$

So $\mathbf{T_z}$ are sufficient statistics for $\psi(\mathbf{Z})$.

Now if finite mixtures of $f(\mathbf{y};\boldsymbol{\lambda})$ are identifiable, then Eq. 5 implies that $p(\boldsymbol{\lambda}) = q(\boldsymbol{\lambda})$. Therefore:

$$
\begin{aligned}
\mathcal{I}(\mathbf{X};\mathbf{T_z}(\mathbf{Y})) &\leq \mathcal{I}(\mathbf{X};\mathbf{Y}) && \text{data-processing inequality} \\
&\leq \mathcal{I}(\phi(\mathbf{X});\mathbf{Y}) && \mathbf{X} \rightarrow \phi(\mathbf{X}) \rightarrow \mathbf{Y}, \text{D.P.I.} \\
&= \mathcal{I}(\psi(\mathbf{Z});\mathbf{Y}) && p(\boldsymbol{\lambda}) = q(\boldsymbol{\lambda}),\ \text{Eq. 3} \\
&= \mathcal{I}(\psi(\mathbf{Z});\mathbf{T_z}(\mathbf{Y})) && \text{Eq. 6} \\
&= \mathcal{I}(\phi(\mathbf{X});\mathbf{T_z}(\mathbf{Y})) && p(\boldsymbol{\lambda}) = q(\boldsymbol{\lambda}),\ \text{Eq. 3} \\
&\leq \mathcal{I}(\mathbf{X};\mathbf{T_z}(\mathbf{Y})) && \text{data-processing inequality} \\
\implies \mathcal{I}(\mathbf{X};\mathbf{T_z}(\mathbf{Y})) &= \mathcal{I}(\mathbf{X};\mathbf{Y}),
\end{aligned}
\tag{7}
$$

which is what we set out to prove. (This last progression is illustrated in Fig. 2B.)

## 4 Relationship to empirical findings

The use of density-estimation algorithms for multisensory integration appears in [2, 15, 16], and in [2], the connection between generic latent-variable density estimation and multisensory integration was made, although the result was shown only empirically. We therefore relate those results to the foregoing proof.

### 4.1 A density estimator for multisensory integration

In [2], an exponential-family harmonium (model distribution, $q$, Fig. 3B) with Poisson visible units ($\mathbf{Y}$) and Bernoulli hiddens units ($\mathbf{Z}$) was trained on simulated populations of neurons encoding arm configuration in two-dimensional space (Fig. 3). An EFH is parameterized by the matrix of connection strengths between units (weights, $W$) and the unit biases, $b_i$. The nonlinearities for Bernoulli and Poisson units are logistic and exponential, respectively, corresponding to their inverse "canonical links" [17].

The data for these populations were created by (data distribution, $p$, Fig. 3A):

1. drawing a pair of joint angles ($\theta_1 =$ shoulder, $\theta_2 =$ elbow) from a uniform distribution between the joint limits; drawing two population gains ($g_p, g_v$, the "reliabilities" of the two populations) from uniform distributions over spike counts—hence $\mathbf{x} = [\theta_1, \theta_1, g_p, g_v]$;

2. encoding the joint angles in a set of 2D, Gaussian tuning curves (with maximum height $g_p$) that smoothly tile joint space ("proprioceptive neurons," $\boldsymbol{\lambda}_p$), and encoding the correspond-

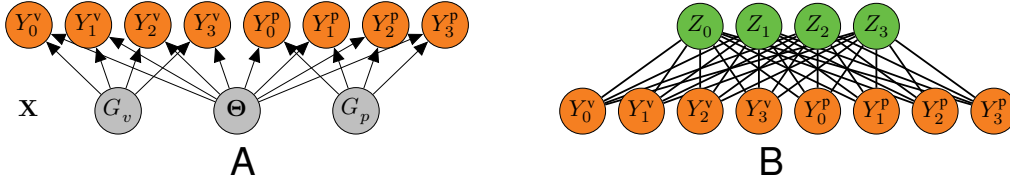

Figure 3: Two probabilistic graphical models for the same data—a specific instance of Fig. 1. Colors are as in Fig. 2. (A) Hand position ($\mathbf{\Theta}$) elicits a response from populations of visual ($\mathbf{Y}^{\text{v}}$) and proprioceptive ($\mathbf{Y}^{\text{p}}$) neurons. The reliability of each population's encoding of hand position varies with their respective gains, $G_v, G_p$. (B) The exponential family harmonium (EFH; see text). After training, an up-pass through the model yields a vector of multisensory (mean) activities ($\mathbf{z}$, hidden units) that contains all the information about $\boldsymbol{\theta}$, $g_v$, and $g_p$ that was in the unisensory populations, $\mathbf{Y}^{\text{v}}$ and $\mathbf{Y}^{\text{p}}$.

ing end-effector position in a set of 2D, Gaussian tuning curves (with maximum height $g_v$) that smoothly tile the reachable workspace ("visual neurons," $\boldsymbol{\lambda}_{\text{v}}$);

3. drawing spike counts, $[\mathbf{y}_{\text{p}}, \mathbf{y}_{\text{v}}]$, from independent Poisson distributions whose means were given by $[\boldsymbol{\lambda}_{\text{p}}, \boldsymbol{\lambda}_{\text{v}}]$.

Thus the distribution of the unisensory spike counts, $\mathbf{Y} = [\mathbf{Y}_{\text{p}}, \mathbf{Y}_{\text{v}}]$, conditioned on hand position, $p(\mathbf{y}|\mathbf{x}) = \prod_i p(y_i|\mathbf{x})$, was a "probabilistic population code," a biologically plausible proposal for how the cortex encodes probability distributions over stimuli [1]. The model was trained using one-step contrastive divergence, a learning procedure that changes weights and biases by descending the approximate gradient of a function that has $q(\mathbf{y}) = p(\mathbf{y})$ as its minimum [18, 19].

It was then shown empirically that the criterion for "optimal multisensory integration" proposed in [1],

$$\Pr[\mathbf{X}|\bar{\mathbf{Z}}] = \Pr[\mathbf{X}|\mathbf{y}_{\text{p}}, \mathbf{y}_{\text{v}}], \tag{8}$$

held approximately for an average, $\bar{\mathbf{Z}}$, of vectors sampled from $q(\mathbf{z}|\mathbf{y})$, and that the match improves as the number of samples grows—i.e., as the sample average $\bar{\mathbf{Z}}$ approaches the expected value $\mathbb{E}_{q(\mathbf{z}|\mathbf{y})}[\mathbf{Z}|\mathbf{y}]$. Since the weight matrix $W$ was "fat," the randomly initialized network was highly unlikely to satisfy Eq. 8 by chance.

## 4.2 Formulating the empirical result in terms of the proof of Section 3

To show that Eq. 8 *must* hold, we first demonstrate its equivalence to Eq. 1. It then suffices, under our proof, to show that the model obeys Eqs. 3 and 5 and that the "mixture model" defined by the true generative process is identifiable.

For sufficiently many samples, $\bar{\mathbf{Z}} \approx \mathbb{E}_{q(\mathbf{z}|\mathbf{y})}[\mathbf{Z}|\mathbf{Y}]$, which is a sufficient statistic for a vector of Bernoulli random variables: $\mathbb{E}_{q(\mathbf{z}|\mathbf{y})}[\mathbf{Z}|\mathbf{Y}] = \mathbf{T}_{\mathbf{z}}(\mathbf{Y})$. This also corresponds to a noiseless "up-pass" through the model, $\mathbf{T}_{\mathbf{z}}(\mathbf{Y}) = \sigma\{W\mathbf{Y} + \mathbf{b}_{\mathbf{z}}\}^2$. And the information about the stimulus reaches the multisensory population, $\mathbf{Z}$, only via the two unisensory populations, $\mathbf{Y}$. Together these imply that Eq. 8 is equivalent to Eq. 1 (see Appendix for proof).

For both the "world" and the model, the function $f(\mathbf{y}; \boldsymbol{\lambda})$ is a product of independent Poissons, whose means $\boldsymbol{\Lambda}$ are given respectively by the embedding of hand position into the tuning curves of the two populations, $\phi(\mathbf{X})$, and by the noiseless "down-pass" through the model, $\exp\{W^{\text{T}}\mathbf{Z} + \mathbf{b}_{\mathbf{y}}\} =: \psi(\mathbf{Z})$. So Eq. 3 is satisfied. Eq. 5 holds because the EFH was trained as a density estimator, and because the mixture may be treated as finite. (Although hand positions were drawn from a continuous uniform distribution, the number of mixing components in the data distribution is limited to the number of training samples. For the model in [2], this was less than a million. For comparison, the harmonium had $2^{900}$ mixture weights at its disposal.) Finally, the noise model is factorial:

$f(\mathbf{y}; \boldsymbol{\lambda}) = \prod_i f(y_i; \lambda_i)$. The class of mixtures of factorial distributions, $f(\mathbf{y}; \boldsymbol{\lambda})$, is identifiable just in case the class of mixtures of $f(y_i; \lambda_i)$ is identifiable [14]; and mixtures of (univariate) Poisson conditionals are themselves identifiable [12]. Thus the mixture used in [2] is indeed identifiable.

## 5   Conclusions

We have traced an analytical connection from psychophysical results in monkey and man to a broad class of machine-learning algorithms, namely, density estimation in latent-variable models. In particular, behavioral studies of multisensory integration have shown that primates estimate stimulus properties with the peak of the posterior distribution over the stimulus, conditioned on the two unisensory cues [3, 4]. This can be seen as a special case of a more general "optimal" computation, viz., computing and representing the entire posterior distribution [1, 6]; or, put differently, finding transformations of multiple unisensory representations into a multisensory representation that retains all the original information about the underlying stimulus. It has been shown that this computation can be learned with algorithms that implement forms of latent-variable density estimation [15, 16]; and, indeed, argued that generic latent-variable density estimators will satisfy the information-retention criterion [2]. We have provided an analytical proof that this is the case, at least for certain classes of models (including the ones in [2]).

What about distributions $f(\mathbf{y}; \boldsymbol{\lambda})$ other than products of Poissons? Identifiability results, which we have relied on here, appear to be the norm for finite mixtures; [12] summarizes the "overall picture" thus: "[A]part from special cases with finite samples spaces [like binomials] or very special simple density functions [like the continuous uniform distribution], identifiability of classes of *finite* mixtures is generally assured." Thus the results apply to a broad set of density-estimation models and their equivalent neural networks.

Interestingly, this excludes Bernoulli random variables, and therefore the mixture model defined by restricted Boltzmann machines (RBMs). Such mixtures are not strictly identifiable [12], meaning there is more than one set of mixture weights that will produce the observed marginal distribution. Hence the guarantee proved in Section 3 does not hold. On the other hand, the proof provides only sufficient, not necessary conditions, so some guarantee of information retention is not ruled out. And indeed, a relaxation of the identifiability criterion to exclude sets of measure zero has recently been shown to apply to certain classes of mixtures of Bernoullis [21].

The information-retention criterion applies more broadly than multisensory integration; it is generally desirable. It is not, presumably, sufficient: the task of the cortex is not merely to pass information on unmolested from one point to another. On the other hand, the task of integrating data from multiple sources without losing information about the underlying cause of those data has broad application: it applies, for example, to the data provided by spatially distant photoreceptors that are reporting the edge of a single underlying object. Whether the criterion can be satisfied in this and other cases depends both on the brain's generative model and on the true generative process by which the stimulus is encoded in neurons.

The proof was derived for sufficient statistics rather than the neural responses themselves, but this limitation can be overcome at the cost of time (by collecting or averaging repeated samples of neural responses) or of space (by having a hidden vector long enough to contain most of the information even in the presence of noise).

Finally, the result was derived for "completed" density estimation, $q(\mathbf{y}) = p(\mathbf{y})$. This is a strong limitation; one would prefer to know how approximate completion of learning, $q(\mathbf{y}) \approx p(\mathbf{y})$, affects the guarantee, i.e., how robust it is. In [2], for example, Eq. 2 was never directly verified, and in fact one-step contrastive divergence (the training rule used) has suboptimal properties for building a good generative model [22] And although the sufficient conditions supplied by the proof apply to a broad class of models, it would also be useful to know necessary conditions.

**Acknowledgments**

JGM thanks Matthew Fellows, Maria Dadarlat, Clay Campaigne, and Ben Dichter for useful conversations.

## Footnotes

[1]An EFH is a two layer Markov random field, with full interlayer connectivity and no intralayer connectivity, and in which the conditional distributions of the visible layer given the hiddens and vice versa belong to exponential families of probability distributions [5]. The restricted Boltzmann machine is therefore the special case of Bernoulli hiddens and Bernoulli visibles.

[2]That the vector of means alone and not higher-order cumulants suffices reflects the fact that the sufficient statistics can be written as linear functions of $\mathbf{Y}$—in this case, $W\mathbf{Y}$, with $W$ the weight matrix—which is arguably a generically desirable property for neurons [20].

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
