[Supplementary Material]

**Appendix to _Sensory Integration and Density Estimation_**

Figure 1: Two Markov chains. (A) $\mathbf{X}$ and $\mathbf{T}$ are conditionally independent given $\mathbf{Y}$. (B) $\mathbf{X}$ and $\mathbf{Y}$ are conditionally independent given $\mathbf{T}$.

**A theorem relating posterior distributions and mutual information.**
Suppose (premises) that $\mathbf{t} = f(\mathbf{y})$, not necessarily invertible, and $\mathbf{y} \sim p(\mathbf{y}|\mathbf{x})$. Then:

$$\mathcal{I}(\mathbf{X};\mathbf{Y}) = \mathcal{I}(\mathbf{X};\mathbf{T}) \iff \Pr[\mathbf{X}|\mathbf{Y}] = \Pr[\mathbf{X}|\mathbf{T}]. \tag{1}$$

_Proof._

$$\begin{aligned}
\mathcal{I}(\mathbf{X};\mathbf{Y},\mathbf{T}) &= \mathcal{I}(\mathbf{X};\mathbf{T}) + \mathcal{I}(\mathbf{X};\mathbf{Y}|\mathbf{T}) && \text{(chain rule of mutual information)}\\
&= \mathcal{I}(\mathbf{X};\mathbf{Y}) + \mathcal{I}(\mathbf{X};\mathbf{T}|\mathbf{Y}) && \text{(chain rule of mutual information).}
\end{aligned}$$

Now the premises imply that $\mathbf{X} \to \mathbf{Y} \to \mathbf{T}$ (a Markov chain; Fig. 1A), so $\mathcal{I}(\mathbf{X};\mathbf{T}|\mathbf{Y}) = 0$. That means that

$$\mathcal{I}(\mathbf{X};\mathbf{T}) = \mathcal{I}(\mathbf{X};\mathbf{Y}) \iff \mathcal{I}(\mathbf{X};\mathbf{Y}|\mathbf{T}) = 0.$$

But:

$$\mathcal{I}(\mathbf{X};\mathbf{Y}|\mathbf{T}) = 0 \iff \mathbf{X} \to \mathbf{T} \to \mathbf{Y} \quad \text{(a Markov chain; Fig. 1B).}$$

The two Markov chains (Fig. 1) are equivalent to:

$$\begin{aligned}
\Pr[\mathbf{X}|\mathbf{Y},\mathbf{T}] &= \Pr[\mathbf{X}|\mathbf{Y}],\\
\Pr[\mathbf{X}|\mathbf{Y},\mathbf{T}] &= \Pr[\mathbf{X}|\mathbf{T}].
\end{aligned}$$

Since the premises imply that the first equality always holds, we have

$$\mathcal{I}(\mathbf{X};\mathbf{Y}|\mathbf{T}) = 0 \iff \Pr[\mathbf{X}|\mathbf{Y}] = \Pr[\mathbf{X}|\mathbf{T}].$$

Stringing the biconditionals together gives the theorem. $\qquad\square$