[Reviews · NeurIPS 2014]

Submitted by Assigned_Reviewer_13

Summary: The paper is motivated by a recent Plos CB paper by Makin et al 2013, which used numerical simulations of a particular latent variable model that (in this model), performing density estimation of sensory population activity yielded latent variables which retained all the information about the underlying stimulus-variable. In the present paper, the authors ask the question of whether these results hold more gernally. They prove mathematically (assuming some sufficient conditions) that this results holds for a range of models, but also provide counter-examples that it does not always hold.
Quality: Results seem sound, but I did not check the proof very rigorously.
Clarity: The structure of the paper could have improved-- the story of the paper is developed slowly and in almost a 'chatty' manner, with definitions/examples and counter-examples slowly leading up to the proof on the main result. On the one hand, this does make for some accessible reading and nicely motivates the particular set of assumptions, on the other hand, it does mean that e.g. the assumptions of the main proof are scattered across the paper. In the end, it is not really clear which models would or would not satisfy the assumptions of the central theorem. Section 4 and first paragraph of 3.1. are a bit confusing.
Originality: To my knowledge, the proof and question are original.
Significance: It is a bit unclear to me whether the results would be or broad interest to the NIPS community, or whether it scope will remain limited to being seen as a 'follow up/reply to' the Makin et al paper.

Summary: A theoretical investigation of the relationship between multisensory integration and density estimation which treats and evaluates previous work on this question more rigorously. Results are interesting, broad significance unclear.

Submitted by Assigned_Reviewer_45

While the relation of sensory integration and density estimation has more or less explicitly been considered many times before, to my knowledge this paper presents a new mathematical framework for formalizing the intuitive connection between these two concepts. The work appears to be solid and provides a proof for its fundamental theorem (Eg. 1). On the other hand, the framework is rather formal and it did not really become clear to me, what we learn from it about the brain. Apparently it is, however, useful for judging previous less formal models, and the authors show that the information-preservation criterion indeed holds for the one in their ref [2].
Summary: The common ground of sensory integration and density estimation is formalized and an information-preservation criterion is proven. The method is useful for judging models of sensory integration.

Submitted by Assigned_Reviewer_46

This work investigates a generative model with latent variables. This model is assumed to perfectly generate some observed distribution, which was originally generated from some other "true" model with latent variables. Specifically, this work finds sufficient conditions so that any reasonable inference procedure done to infer the latent variables in the generative model from the observed variables would retain the information contained in these observed variables about the "true" latent variables.

Quality: The core derivations (section 3.2+appendix) appear to be correct.

Clarity: The organization of this manuscript seems rather fragmented and confusing, for a purely theoretical paper. It would benefit from clearly and explicitly stating in the beginning: (1) the general probabilistic model (2) all the assumptions, and (3) the main result. Also, some parts could be significantly shortened or removed (e.g. the first few paragraphs of section 2.1).

Originality: As far as I am aware, these results are novel.

Significance: It is not very clear to me how strong are these results. Most importantly, The assumptions of perfect observed density reconstruction, and of an identical noise model seem rather strong and limiting.

Minor comments:
1)Something seems to be missing from Eq. 3 ( \in what set?).
2)Line 240 EHF->EFH

%%% Edited after author feedback %%%
Following the author response, I now agree that the "identical noise model" assumption is not as weak as I thought. This is just equivalent to the assumption that X "the state of the world" is deterministically converted into the direct input to the sensory neurons. Though this is probably not true, I agree that this can be considered as a reasonable first approximation.
The relaxation of the other assumption ("perfect observed density reconstruction") is not yet clear to me. The authors should clarify this important point in the final version of the paper.

Summary: This paper is technically sound, and as far as I know, novel. It could use some more work to improve its clarity and organization, and clarify its significance.
Author Feedback
Author rebuttal: The referees appear to have understood the paper and what it purports to accomplish. We would only add a few comments on significance and scope.

1. Significance and interest to the broader NIPS community. The idea that learning in the cortex should be viewed as a form of latent-variable density estimation is widespread in computational neuroscience. Yet it is clear that whatever generative models are instantiated by neural circuits, they are not identical to the true generative processes of the world; so the relationship between inference in these trained networks--their ultimate purpose--and inference to states of affairs in the world is not in fact obvious. Does the former impair the latter, by throwing away stimulus information? As the counter-examples in our paper show, successful density estimation alone can't rule this out. Our proof supplies such a guarantee for a class of models, and provides a framework for its extension to other classes.

We think the relationship with sensory integration is at least as important (the construction of the paper reflects this), and here the result is also conceptual: Multisensory integration is a topic of long-standing interest in the neuroscience and cognitive-science communities; and density estimation is a well understood approach in machine learning and statistics. Tying sensory integration to density estimation embeds it in a broader context, furthering (we think) our understanding of it: Sensory integration is just a necessary byproduct, for a large class of models, of the sensible goal of building good models of the incoming data. References (2) and (15) in our paper make a similar point, but the empirical/numerical nature of their results limit (in our view) their implications: it was not obvious how well those results generalized.

2. What does this tell us about the brain? It certainly suggests that "higher" sensory areas--like posterior parietal cortex--might operate on the same learning principles as primary areas--like auditory (Lewicki et alia) and visual (Olshausen et alia) cortex--just on different data. (When the stimulus parameter is location, this could be summarized as doing for the "where" pathway what has been done for the "what" pathway.) The proof also widens the class of density-estimation algorithms that could be implemented by the cortex to produce the observed psychophysical results (referenced in section 2.1).

3. Restrictiveness of assumptions. Our stipulation that the "noise models match" is not as strong as it perhaps appears: it just says that whether a neuron is driven by afferents or by feedback should make no difference to the noise model (e.g., the neuron might emit Poisson-distributed spike counts). This is the default assumption in neural networks.

The stipulation of perfect observed-density reconstruction is strong, as we note at the end of the paper. We hope that the reviewers agree, however, that our result is quite useful as it stands, and that where it is weak, it points the way toward construction of a stronger version. [To cover cases of approximate density estimation, where KL{p(y)|q(y)} < epsilon, identifiability would be used to put bounds on KL{p(lambda)|q(lambda)}, which in turn would bound the information loss in lines 3 and 5 of Eq'ns 8, which would bound the overall information loss.]

4. Model/assumptions/result in one place. We are happy make these changes to the paper. Briefly, here:

(a) The general probabilistic model. This is given by the graphical models in Fig. 1. "The world" generates data according to the model on the left ("data distribution"), and "the brain" uses the model on the right. None of the distributions labeled in the diagram need be equal to each other, and in fact X and Z may have different support.

(b) The assumptions.
i. The noise models "match": Eq'n 3 (see also comment (3) above).
ii. The number of hidden-variable states is finite, but otherwise arbitrarily large.
iii. Density estimation has been successful; i.e. the data and model marginals over Y match: Eq'n 2.
iv. The noise model/conditional distribution f(y;lambda) is identifiable: if p(y) = q(y), then p(lambda) = q(lambda). This condition holds for a very broad class of distributions (see the top of page 6 and reference 12).

(c) The main result. Information about the stimulus is retained in inferring the latent variables of the model, i.e. in the "feedforward" (Y to Z) pass of the model. More precisely, information loss is due only to noise in the hidden layer (which is unavoidable), not to a failure of the inference procedure: Eq'n 1.

5. Minor issues. We meant the backward epsilon to be read "such that."